# Performance Assessment of a Commercial Continuous-Wave Near-Infrared Spectroscopy Tissue Oximeter for Suitability for Use in an International, Multi-Center Clinical Trial

**DOI:** 10.3390/s21216957

**Published:** 2021-10-20

**Authors:** Lorenzo Cortese, Marta Zanoletti, Umut Karadeniz, Marco Pagliazzi, M. Atif Yaqub, David R. Busch, Jaume Mesquida, Turgut Durduran

**Affiliations:** 1ICFO-Institut de Ciències Fotòniques, The Barcelona Institute of Science and Technology, 08860 Castelldefels, Spain; marta.zanoletti@icfo.eu (M.Z.); umut.karadeniz@icfo.eu (U.K.); marco.pagliazzi@yahoo.it (M.P.); atif.yaqub@icfo.eu (M.A.Y.); turgut.durduran@icfo.eu (T.D.); 2Department of Anesthesiology and Pain Management and Department of Neurology, University of Texas Southwestern Medical Center, Dallas, TX 75390, USA; david.busch@utsouthwestern.edu; 3Àrea de Crítics, Parc Taulí Hospital Universitari, 08208 Sabadell, Spain; jmesquida@tauli.cat; 4Institució Catalana de Recerca i Estudis Avançats (ICREA), 08015 Barcelona, Spain

**Keywords:** continuous wave near-infrared spectroscopy, vascular occlusion test, local tissue oxygenation, multi-center clinical trial, medical optics, light propagation in tissue

## Abstract

Despite the wide range of clinical and research applications, the reliability of the absolute oxygenation measurements of continuous wave near-infrared spectroscopy sensors is often questioned, partially due to issues of standardization. In this study, we have compared the performances of 13 units of a continuous wave near-infrared spectroscopy device (PortaMon, Artinis Medical Systems, NL) to test their suitability for being used in the HEMOCOVID-19 clinical trial in 10 medical centers around the world. Detailed phantom and in vivo tests were employed to measure the precision and reproducibility of measurements of local blood oxygen saturation and total hemoglobin concentration under different conditions: for different devices used, different operators, for probe repositioning over the same location, and over time (hours/days/months). We have detected systematic differences between devices when measuring phantoms (inter-device variability, <4%), which were larger than the intra-device variability (<1%). This intrinsic variability is in addition to the variability during in vivo measurements on the forearm muscle resulting from errors in probe positioning and intrinsic physiological noise (<9%), which was also larger than the inter-device differences (<3%) during the same test. Lastly, we have tested the reproducibility of the protocol of the HEMOCOVID-19 clinical trial; that is, forearm muscle oxygenation monitoring during vascular occlusion tests over days. Overall, our conclusion is that these devices can be used in multi-center trials but care must be taken to characterize, follow-up, and statistically account for inter-device variability.

## 1. Introduction

Continuous-wave near-infrared spectroscopy (CW-NIRS) [1,2] is a non-invasive optical technique that allows the direct determination of local tissue oxy- and deoxy-hemoglobin concentrations at the microvascular level. An important challenge in interpreting the data generated by the rapidly expanding number of CW-NIRS devices offered for use in the clinical environment is the unknown reproducibility of the measurements and the inherent variability between devices. Past studies have questioned the reliability of absolute oxygenation measurements and highlighted differences between different devices and brands, hence highlighting the need for improved standardization [1,3,4,5,6,7].

Despite this variability, the use of CW-NIRS sensors for tissue oxygenation monitoring is well accepted and established in both clinical and research settings [1,2,8,9]. This widespread use is driven by the advantages of CW-NIRS technology; i.e. the measurements are non-invasive and the devices are compact, portable, and relatively low-cost. The best established clinical uses of CW-NIRS are in the intraoperative detection of cerebral ischemia [10,11,12] and in the cerebral and peripheral muscle hemodynamic monitoring of critically ill patients in the intensive care unit (ICU) [13,14,15,16,17,18,19].

During the first peak of the COVID-19 pandemic (April 2020), ICFO—the Institute of Photonic Sciences (Spain)—and Hospital Parc Taulí de Sabadell (Spain) built upon the promising results of NIRS studies on critically ill patients undergoing mechanical ventilation for acute respiratory distress syndrome (ARDS) [18] to establish a study of the endothelial and microvascular health of COVID-19 patients undergoing intensive care by monitoring the local muscle hemodynamics through NIRS [15,20]. This study grew into a multi-center clinical campaign, which is currently being run simultaneously in 10 hospitals worldwide (HEMOCOVID-19 project [21]; see following section).

Given the exceptional situation during the first pandemic peak and the necessity to rapidly begin this clinical study, we have opted to use commercially available, research-grade CW-NIRS devices (PortaMon by Artinis Medical Systems, NL). The choice of PortaMon was dictated by the fact that a stock of these devices was immediately available from the manufacturer and was also within the project budget. In addition, the devices have characteristics that are particularly useful for this study under protocols to prevent COVID-19 transmission; i.e., they are battery-operated, wireless and remotely controlled (e.g., from outside an isolation room), easy-to-operate, disinfectable, and suitable for measuring the hemodynamics of the *brachioradialis* muscle of the forearm (critical to implement the HEMOCOVID-19 measurement protocol).

Despite being commercially available and having been used in several research studies, with particular success in the sport/athletics field [22,23,24,25,26,27], PortaMon is not certified as a medical device. For this reason, and as a result of the above-mentioned standardization issues of CW-NIRS, we have opted to carry on extensive tests on phantoms and in vivo. Moreover, a multi-center experimental study itself results in a number of issues regarding the comparison of measurements performed in different conditions, by different operators and longitudinally over several months of a clinical campaign [28]. These issues drove our work to accurately assess and compare the performances of these devices, as required by the structure of the HEMOCOVID-19 clinical campaign.

In this paper, we present how we have addressed the above-mentioned challenges by characterizing and comparing the performance of all the devices used for the HEMOCOVID-19 campaign using tissue-stimulating phantom and in vivo experiments on the forearm muscle of healthy subjects (see the schematics of challenges and tests in Figure 1). The aim is the assessment of the performance of the devices in terms of measurement precision and reproducibility and the quantification of any differences between devices. We have tested the precision of single acquisitions, the reproducibility of probe repositioning in the same position, measurement reproducibility over time—that is, after hours, days, and months—and the reproducibility due to different operators. We have also characterized a set of tissue-simulating phantoms to be used for assessing the day-by-day reliability of each device, and we have tested the reproducibility of the protocol of the HEMOCOVID-19 clinical trial; that is, forearm muscle oxygenation monitoring during a vascular occlusion test.

## 2. HEMOCOVID-19 Clinical Trials

HEMOCOVID-19 [21] is a multi-center, international clinical research project (ClinicalTrials.gov identifier NCT04689477 and NCT04692129) that aims to use continuous-wave near-infrared spectroscopy devices in intensive care units to aid in the clinical management of severely ill COVID-19 patients at multiple stages, supporting a continuum of care.

The objective of the study is to assess the endothelial health of critically ill COVID-19 patients with the long-term aim of providing doctors with new prognostic biomarkers based on minimally invasive optical measurements. The endothelial health of the subjects is assessed by the dynamic CW-NIRS measurement of local tissue oxygen saturation and the total hemoglobin concentration of the forearm muscle during the three phases of a vascular occlusion test—baseline, ischemic occlusion, and recovery after releasing the occlusion [15]. In addition, as a sub-study, the tissue oxygen saturation and total hemoglobin concentration of a small group of mechanically ventilated patients was measured before and after transitioning a patient from a supine to prone position to assess the effectiveness of this rescue strategy on the local tissue oxygenation.

The project was coordinated by ICFO—the Institute of Photonic Sciences (Spain)—and Hospital Parc Taulí de Sabadell (Spain) and in the first phase included eight hospitals in four different countries (Spain, Mexico, the USA, and Brazil). Currently, the HEMOCOVID-19 project includes 10 hospitals recruiting patients in 5 different countries [21].

## 3. Materials and Methods

### 3.1. CW-NIRS Devices

For the initial phase of the HEMOCOVID-19 clinical trial (see Section 2), 10 devices were required to launch the project in 8 clinical centers, and 2 devices were kept in-house at ICFO for continued quality control testing. Additional partners have joined the consortium since then. The following characteristics were required when selecting the devices for use:Readily available for delivery within 30 days;Total cost within the limited project budget;Should provide both the trends and an estimate of the absolute value of the blood oxygen saturation;Should be suitable for use at an intensive care unit with regard to the restrictions introduced by the COVID-19 pandemic, including features such as the following:–Wireless/remote controlled;–Disinfectable (with alcohol) between patients;–Easy-to-operate with remote-training only;No disposable parts;–Can be re-utilized without leaving the containment zone;Minimal footprint in contact with the tissue;–Should come with customer support directly from the company during the pandemic lock-downs.Should be suitable to use at the measurement site—the *brachioradialis* muscle of the forearm.

These constraints made instrument selection a challenge. For example, none of the devices in the market with a medical-device authorization (CE, FDA, or other equivalent approvals) met these requirements. Eventually, we decided to utilize PortaMon (Artinis Medical Systems, NL) [30] devices for the clinical trial. This device found particular success during recent years and has been validated and extensively used, primarily in the sport/athletic field, to monitor local muscle oxygenation [4,22,23,24,25,26,27].

Briefly, PortaMon is a portable, wireless, battery-operated CW-NIRS system consisting of three couples of light emitting diodes (LED) as sources at nominal wavelengths of 760 and 850 nm at different distances from the receiver (30, 35, and 40 mm). It is capable of continuous monitoring with a temporal resolution of 0.1 s, reporting the local tissue oxygen saturation index (TSI), oxy- and deoxy-hemoglobin concentrations (respectively, $HbO_{2}$ and Hb), and the total hemoglobin concentration (THC). TSI is an index proportional to the more commonly utilized $StO_{2}$. The absolute values are retrieved by using so-called spatially resolved spectroscopy and the modified Beer–Lambert law [2,4,8,31].

After the initial tests were conducted, we acquired 13 (with 1 device as an additional back-up) devices. Two additional devices were loaned by the manufacturer and were added to the tests after the initial order since (as described below) significant inter-device variability was found, with some units systematically out of the empirically defined acceptance range. In this manuscript, the devices are identified by their “id” number (id36, id38, id40, etc.). A photo of two devices is reported in Figure 2a, showing both top and bottom views.

### 3.2. Tissue-Simulating Phantoms, Type 1

In our laboratory, a set of commercial tissue-simulating, homogeneous, solid phantoms (Biomimic Optical Phantoms, INO, Québec, Canada) was available. This type of phantoms is commonly utilized for testing diffuse optical devices as they provide a relatively well established prior knowledge of the optical properties, good homogeneity, and stability over long periods (years) [32,33].

For this part of the study, we utilized two phantoms (Type 1 phantoms): (1) Biomimic PB300 (INO PB300) with nominal values of reduced scattering coefficient $\mu_{s}^{\prime }=10\;$ cm${}^{-1}$ and absorption coefficient $\mu_{a}=0.1\;$ cm${}^{-1}$ at 785 nm, (2) Biomimic PB312 (INO PB312) $\mu_{s}^{\prime }=5.3\;$ cm ${}^{-1}$ and $\mu_{a}=0.14\;$ cm ${}^{-1}$ at 785 nm. For all the measurements, the phantoms were prepared with a custom mask, which assists in the repeatable and reliable placement of the device on the phantom surface (see Figure 2b).

We note here that the wavelength dependence of the optical properties of these phantoms were *not* tuned to provide a specific TSI or THC equivalent value. Since our goal was not to validate the absolute values of these parameters with these phantoms, we took an approach whereby we used the mean value from all devices as the expected nominal value. Two phantoms were utilized to provide two different measured light levels, partially testing the devices’ dynamic range.

**Figure 2 sensors-21-06957-f002:**
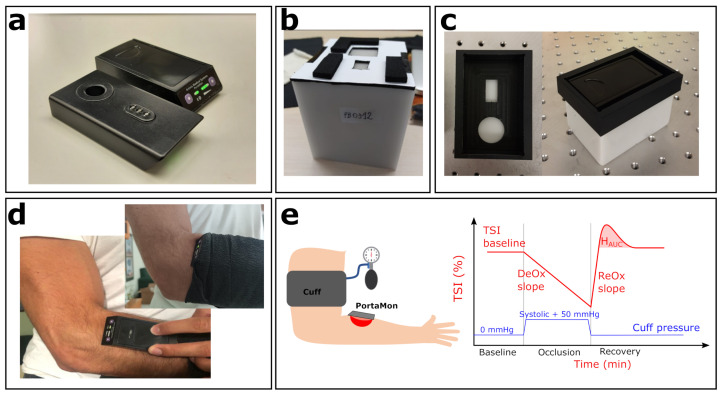
Experimental setup. (**a**) Two PortaMon (Artinis Medical Systems) devices used in this study; top and bottom view. (**b**) Biomimic PB312 phantom, with the custom mask for PortaMon placement. (**c**) One of the BioPixS phantoms, together with the custom mask (left panel, top view) and the PortaMon placed (right panel). (**d**) PortaMon placement for forearm muscle measurements. (**e**) Sketch of the VOT procedure together with a visual representation of the measured relevant parameters.

### 3.3. Tissue-Simulating Phantoms, Type 2

After the initial tests, we observed that there was a systematic variability between different devices. This difference, as discussed below, is quite minimal, but since it has been detected and as its dependence on environmental conditions, the age of the device, hours of use, and other factors is unknown, we sought a solution that would involve utilizing identical phantoms alongside each system. As previously discussed, this is not a trivial problem since the manufacturing, characterization, and maintenance of such phantoms is a complex matter that is still being tackled by both academia and the industry [28,33,34].

A further consideration was the availability of such phantoms at short notice and their cost-effectiveness. The most suitable candidate was identified to be the devices produced by BioPixS (Cork, Ireland, www.biopixstandards.com, acessed on 6 September 2021). Ten (nominally) identical phantoms (BioPixS-Matrix-CCB5d, Type 2) from the same class of solid phantoms as Type 1 (see above) phantoms (8.5×6×4.5 cm ) that were manufactured from the same batch of materials with a nominal reduced scattering coefficient $\mu_{s}^{\prime }=10$ cm${}^{-1}$ and absorption coefficient $\mu_{a}=0.1$ cm${}^{-1}$ at 740 nm were produced and characterized by the manufacturer. The declared inter-phantom difference is 0.8% for the absorption coefficient and 1.4% for the scattering coefficient at 740 nm.

As for the Type 1 phantoms, we prepared a custom mask for reliable positioning on Type 2 phantoms as shown in Figure 2c.

### 3.4. In Vivo Measurements

All the in vivo studies were conducted according to the guidelines of the Declaration of Helsinki and approved by the local Ethics Committee. Subjects were asked to provide informed consent.

In these protocols, the goal was to evaluate the repeatability and variability during the resting condition for each device and also the variability of the vascular occlusion test (VOT) with a single device. To evaluate the repeatability and variability during the resting condition, the forearm *brachioradialis* muscle of the same subject (age: 26; gender: male) was measured. To reduce the variability to physiological changes in the muscle, the subject was at rest sitting on a chair, with the arm resting in a stable position on the arm of the chair.

The vascular occlusion test protocol consisted of continuously monitoring the TSI during a baseline period of three minutes, a period of complete arterial occlusion of three minutes, and a period of recovery of five minutes. Arterial occlusion—i.e., ischemia—was induced by inflating a typical arm blood pressure cuff placed on the biceps at a pressure of 50 mmHg above the systolic pressure of the subject. This protocol was repeated on a separate subject for 20 different days during the same month, on the same healthy subject (age: 41; gender: male) in the supine position. A sketch of the measurement procedure is reported in Figure 2e together with the visual representation of the relevant parameters that were evaluated: TSI baseline, deoxygenation slope—DeOx, reoxygenation slope—ReOx, and hyperemic response—$H_{AUC}$. $H_{AUC}$ is the area under the hyperemic peak. DeOx is calculated by linearly fitting the first minute of the curve TSI vs. Time during the occlusion period. ReOx is calculated by linearly fitting the same curve from the instant the occlusion is released up to the instant the TSI returns back to its baseline value.

### 3.5. Description of Tests

A summary of all the tests performed is reported in Table 1. As mentioned above, the aim of these tests was to assess the performance of the devices in terms of the variability and reproducibility of the measurements when performed under different conditions. The tests were devised to match the overall need to acquire data from multiple locations with different devices, by different operators and over a minimum time-span of a year. Further considerations included that fact that the tissue of interest—a muscle—is soft; therefore, the hemodynamics are affected by the probe pressure but cannot be controlled with a quantitative feedback mechanism in this particular case. In the following sections, we describe each test in detail. Table 1 uses capital letters as an identifier for each test.

We began by evaluating the the warm-up time of the device (Test *A*) and stability of TSI and THC over eight hours of continuous acquisition starting immediately after turning the device on. Type 1 phantoms were utilized in this test and three different devices were utilized. We opted not to use the whole set of devices since the results from the first three were quite similar and due to the urgency of starting the clinical trial. The goal of this test was to provide instructions for the user about the use of the device. It was important to identify this point and if there was any variability between different devices, since, as a battery-operated device, it could not be switched on continuously. The warm-up time was evaluated by assuming that the measured TSI and THC would stabilize around a mean value once the device was ready for use.

All the subsequent tests (*B* to *F*) also used Type 1 phantoms where each measurement consisted of five subsequent acquisitions of 20 s each, after removing and rapidly replacing the device in position. This measurement was repeated for different devices, by different operators, and in different periods (different days, different months). In addition, to test the variability at different times of the same day, we repeated one measurement with one device at the beginning and at the end of the same day.

Furthermore, as indicated in Table 1, several of these tests (B,C,F) were also performed in vivo on the *brachioradialis* muscle, in the forearm, as detailed above. Each device was placed carefully, as would be done during the clinical trial, and covered with a black bandage to avoid background light, as shown in Figure 2d. For each device, four subsequent single acquisitions of 20 s each were repeated by rapidly removing and replacing the device in the same position. This procedure was repeated two times (two sets of four subsequent acquisitions per device) by randomizing the order of the devices. This precaution was taken in order to avoid that changes of muscle hemodynamics over time would not impact the same device twice in the same manner.

We also characterized (Test *G*) a set of 10 phantoms of Type 2 (see above) to evaluate their potential to be utilized with every device for on-site quality control on a day-to-day basis. These phantoms were characterized with one PortaMon (id50) on three different days. Each day, five subsequent acquisitions of 20 s each were performed after removing and replacing the device rapidly in the same position.

Lastly, we tested (Test *H*) the reproducibility of VOT; that, is the measurement protocol of the HEMOCOVID-19 clinical trial as described above. This protocol was repeated with a single device (id63), once per day, for 20 different days of the same month, on a single healthy subject laying supine for at least 30 min prior to measurement. Please note that this device (id63) was acquired independently by a HEMOCOVID-19 partner and is not reported in the other tests.

For all the tests, the variability and reproducibility of a measurement and the differences between devices were evaluated by calculating the coefficient of variation CV, defined as $CV\left[\%\right]=100\cdot \sigma_{x}/\langle x\rangle$, where *x* is the measured quantity (i.e., TSI and THC), $\sigma_{x}$ is the standard deviation, and 〈x〉 is the average.

## 4. Results and Discussion

### 4.1. Phantom Measurements: Stability and Warm-Up Time (Test A )

The results of the stability and warm-up time tests (Test *A*) are reported in Figure 3, for TSI and THC for all three devices (id38, id40 and id50). After turning on a device, we notice a first period of warm-up during which the TSI rapidly increases and THC rapidly decreases towards their average stable values. For all the devices, stability is reached approximately one hour after switching on the device (CV after the first hour to the end of the experiment <0.3% both for TSI and THC). However, this period is prohibitively long for a battery-operated device to be used at an intensive care unit. Further evaluation shows that a quite accurate estimation of the stable values is reached after 10 minutes. At this point, TSI reaches a value less than 2% lower than the stable value and THC reaches a value less than 1% higher than the stable value. Since the main goal of the HEMOCOVID-19 trial is to characterize the VOT-associated dynamics that are described above, we decided to instruct the users to wait approximately 10 min after the device was turned on before starting the data acquisition.

If further accuracy with respect to the absolute values is needed, additional care should be taken. According to the manufacturer, this may be due to the equalization of the temperature in the opto-electronic components. While this was not explicitly tested to avoid contamination due to physiological changes, the manufacturer’s and our previous experience indicates that keeping the device in contact with the body speeds up this process due to the higher body temperature compared to the typical room temperature. An additional potential precaution could be to store the devices above a temperature-controlled phantom during storage close to body temperature.

**Figure 3 sensors-21-06957-f003:**
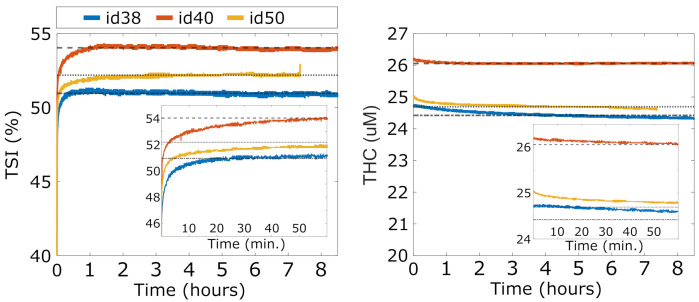
Overnight stability of three devices (id38, id40, and id50). In the inset, zoomed image of the TSI and THC during device warm-up.

### 4.2. Phantom Measurements: Variability and Reproducibility (Test B to Test F)

TSI and THC values related to the two Type 1 phantoms (Biomimic PB300 and PB312) were measured with 13 different devices, and the measurements were repeated multiple times. The results are summarized in Figure 4 together with the average value over all the devices. As detailed above, these measurements allowed us to retrieve the variability of the measured parameters over a single acquisition of 20 s, after the immediate replacement of the device in the same position, and the reproducibility of the measurement after hours, days, and months (as indicated in the figure legends). In addition, we evaluated the reproducibility when the measurement was performed by different operators. Finally, the differences between different devices were measured.

All the results are summarized in Table 2. We note that, in phantoms, the TSI and THC signals are very stable over a single acquisition of 20 s, with a coefficient of variation (CV) of <0.1%. The variability due to device repositioning over the phantom is higher but also very low compared to the expected physiological changes [15,18,20,35] (also in VOT examples below) with a CV of <0.3%. We note that the custom mask allowed the reliable placement/re-placement of the device in the same position, which improved the results. Nevertheless, as expected, the placement/re-placement variability is higher than the static measurements, demonstrating the importance of the careful characterization of these parameters for evaluating the capabilities of these devices. As shown below, the in vivo variability is significantly higher than this value.

The tests also demonstrate a very good reproducibility of the results over hours, days, and months. In all these cases, we obtained comparable variabilities, with a CV of <1.5% both for TSI and THC. A slightly higher variability (CV < 3%) was registered when reproducing the measurements over different months. In addition, the variability due to different operators was comparable to the variabilities due to probe replacement and over different hours, days, and months.

Lastly, we detected significant differences between different devices (CV < 5%) which was significantly larger than all the intra-device CVs in phantoms. These differences can be ascribed to device components (i.e., detector, laser, electronics), their assembly, and the manufacturer’s internal calibration. A careful examination of Figure 4 shows that some devices (e.g., id36 and id48) systematically produced TSI or THC values that were outside one standard deviation of the mean of the whole set of devices; i.e., outside the dashed lines. Even though these values reflect problems with the absolute measurements and do not indicate a problem with the evaluation during the VOT tests, in order to minimize complications in data analysis, we opted to take some actions to minimize the risks. The first action was to discard these devices from the clinical trial. The second was to procure phantoms of Type 2 to evaluate whether it is feasible to keep track of these systematic changes on site and over time for each device. Finally, it was decided that these parameters should be considered as confounders in the statistical analyses of the HEMOCOVID-19 clinical campaign data, whose description is beyond the scope of this paper.

**Figure 4 sensors-21-06957-f004:**
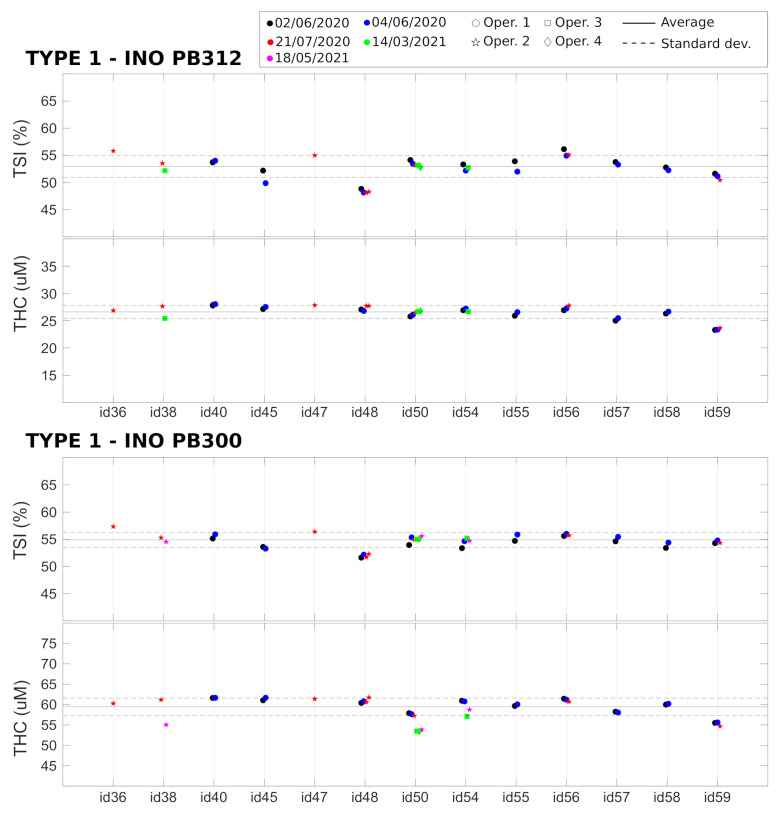
Device characterization of Type 1 phantoms, performed on different days and months by different operators. Horizontal lines represent the average (solid line) ± standard deviation (dashed lines) over all the devices.

These findings have both positive and negative implications for CW-NIRS devices. It is positive that the devices are stable over time and, with the good engineering of the device–phantom interface, could be utilized in a reproducible manner. However, the intra-device variability in absolute values is not characterized in advance, and corrective actions are not implemented. We suggest that all manufacturers take this into account for the quality control of their devices. We note that previous studies have demonstrated significant variability between devices from different manufacturers that affect both the absolute values and the changes in measurements [6,13,36,37]. These findings also highlight the importance of reliable, durable phantom standards in this field.

**Table 2 sensors-21-06957-t002:** Variability of Type 1 phantoms. Apart from the last column, which describes the inter-device variability, all the CVs reported are intra-device variabilities averaged over all the devices. See text for details.

Parameter	Phantom	${\mathrm{CV}}_{\mathrm{single}\;\mathrm{acq}.}$	${\mathrm{CV}}_{\mathrm{replac}.}$	${\mathrm{CV}}_{\mathrm{hours}}$	${\mathrm{CV}}_{\mathrm{days}}$	${\mathrm{CV}}_{\mathrm{months}}$	${\mathrm{CV}}_{\mathrm{operator}}$	${\mathrm{CV}}_{\mathrm{device}}$
		(%)	(%)	(%)	(%)	(%)	(%)	(%)
TSI	PB300	0.08	0.08	0.8	1.1	0.8	0.09	2.5
TSI	PB312	0.09	0.2	0.3	1.3	1.2	0.5	3.8
THC	PB300	0.04	0.2	1.3	0.3	2.8	0.1	3.6
THC	PB312	0.04	0.3	0.09	0.9	2.1	0.2	4.5

### 4.3. Phantom Measurements: Towards On-Site Quality Control

As described before, these findings led us to consider the evaluation of the potential of commercial tissue-simulating phantoms for inclusion in the “kit” that was shipped to each center. These Type 2 phantoms have been characterized (Test *G*) by a single device (id50) whose results are shown in Figure 5 and in Table 3. Different phantoms from the same batch show slightly different TSI (51.7±1.6%,CV=3.0%) and THC (41.9±2.3μ M, CV= 5.5% ). These small differences between phantoms are well within the detectability range of the device, since the CV over the 10 phantoms ($CV_{TSI}<1.6\%$ and $CV_{THC}<2.5\%$) is well above the intra-device CV reported earlier. The average CV between measurement in different days was found to be 1.2% for TSI and 1.4% for THC, similar to the values reported for the measurements on Type 1 phantoms (see Table 2).

These findings led us to conclude that while these phantoms could be useful to evaluate whether the devices degraded or their performance changed over time, they would not be reliable enough as a means to correct for the inter-device systematic differences that were observed from the tests on Type 1 phantoms. Due to the constraints imposed on us by the on-going COVID-19 pandemic, we opted not to distribute these phantoms to individual sites. We are now working with the manufacturer to devise reliable means to produce and evaluate phantoms that are suitable for future use.

However, these findings are also quite encouraging in highlighting the need for professional phantom suppliers and well-defined standards in the field. We believe that on-going efforts by us (e.g., the European Commission Horizon 2020 projects LUCA, VASCOVID, TinyBrains, BitMap) and others (e.g. Acrin trial, SOLUS and NeuroOpt projects, Photonics21 activities) are moving towards establishing robust methods and materials, which are crucial for the clinical adoption of these technologies. We also encourage the device manufacturers to undertake additional efforts to provide this type of characterization information to the users.

**Figure 5 sensors-21-06957-f005:**
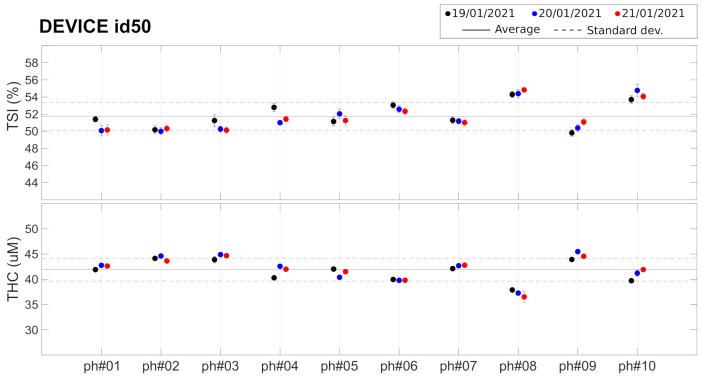
Set of 10 Type 2 phantoms measured with device id50. Horizontal lines represent the average (solid line) ± standard deviation (dashed lines) over all the phantoms. See text for details.

### 4.4. In Vivo Repeatability

The results of the in vivo variability measurements (Tests A,C,F) on the forearm muscle (*brachioradialis* muscle) are reported in Figure 6 and are summarized in Table 4. During a single acquisition of 20 s, the average CVs over all the devices was approximately 0.9% for TSI and 2.4% for THC, which are much larger than the values measured on phantoms (see Section 4.2). This reflects that muscle hemodynamics changed during the 20 s of each acquisition. This may be due to various effects that are beyond the scope of this paper to evaluate. For example, the placement of the device on the arm introduces additional, local pressure, which may lead to the redistribution of blood and changes in the oxygen metabolism.

A higher variability (8.5% for TSI and 17.5% for THC) was observed when replacing the probe in the same position over the forearm muscle. This large variability is presumably due to errors in repositioning the device in the same position with the same probe pressure, in addition to the above-mentioned physiological changes occurring during the period of the measurement session. We note that the measurement sessions were roughly five minutes for a single set of four replacements in a single device, with one hour between the first and second set of four acquisitions. No significant differences were observed between the variability obtained by considering only the first set, only the second set, and when considering both sets of for acquisitions. Therefore, in Table 4, we have reported the value obtained considering all the acquisitions without discriminating between first and second sets. Overall, the results reported here are in line with previous studies and other equivalent CW-NIRS devices [4,6].

These measurements have demonstrated that the differences registered between devices in phantoms (see Table 2) do not affect the reliability of the in vivo measurements, since the in vivo variability due to probe repositioning is much larger (see Table 4). These findings once more highlight the difficulty of obtaining reliable absolute measurements using CW-NIRS devices. This may be remedied by the intensive training of the operators, better engineering of the device–tissue interface (“probe”)—for example, by controlling the applied pressure—better definition of the device placement, and a better understanding of the factors that affect the hemodynamic stability of the underlying muscle. Some of these limitations are well known for CW-NIRS devices, and the on-going efforts being made to reduce the size and cost of time-resolved NIRS systems (TR-NIRS) [13,38,39,40,41] are an important step in this direction.

We also note that HEMOCOVID-19 was undertaken during the height of the pandemic to obtain what we believe is clinically important and urgent information. Previous studies in similar critically ill populations have identified some of these problems and utilized thenar eminence as a measurement location [15,17,42,43,44,45], which was not accessible to us due to the size of these devices. An ongoing project, VASCOVID [41], is attempting to minimize these confounders by incorporating a TR-NIRS system with a small footprint probe with different sensors in a multi-modal platform.

Since the patterns of the dynamics of induced hemodynamic changes are affected in a different manner, we now move onto the final set of in vivo measurements during VOT.

**Figure 6 sensors-21-06957-f006:**
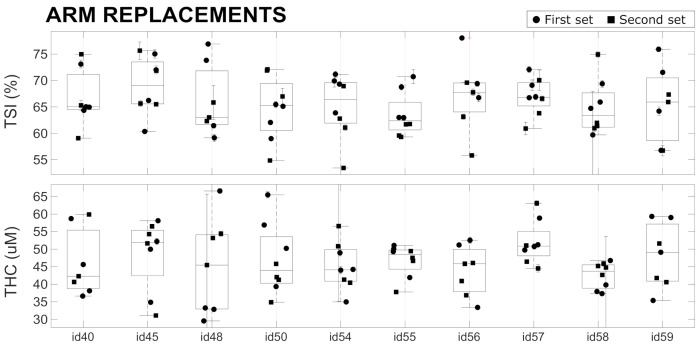
In vivo replacement tests performed with 10 devices on a single healthy subject. See text for details.

### 4.5. In Vivo Characterization of the Repeatability of Dynamics

The results of the repeated VOTs that were performed once per day on 20 different days on the same subject during one month are reported in Figure 7 and summarized in Table 5. Different parameters were extracted from the response to the VOT, as described above, and no significant trends were registered during the 30 days of measurement. Overall, we observe a quite stable baseline level for $TSI_{baseline}$ (CV=4%), while for the other parameters (DeOx, ReOx, $H_{AUC}$ and $THC_{baseline}$), the variability is higher and ranges between 16–27%. This could be expected since the response to VOT is always variable within an individual. This has been discussed previously and was shown to be less variable in other, smaller muscles [42,43]. In addition, the VOT protocol adopted could influence the variability of the retrieved parameters: as discussed in previous studies, maintaining the ischemia up to a pre-determined minimum value of TSI (ischemia-depth-based protocol) should reduce the variations with respect to maintaining the ischemia for a fixed time (ischemia-time-based protocol) [42,45,46].

Lastly, we comment that the individual variabilities reported here allow researchers to discriminate the contrast in VOT-related parameters between healthy controls and severe intensive care unit patients with good and poor outcomes in various situations, such as for ARDS, septic, trauma, and COVID-19 patients [15,18,44,47,48].

**Figure 7 sensors-21-06957-f007:**
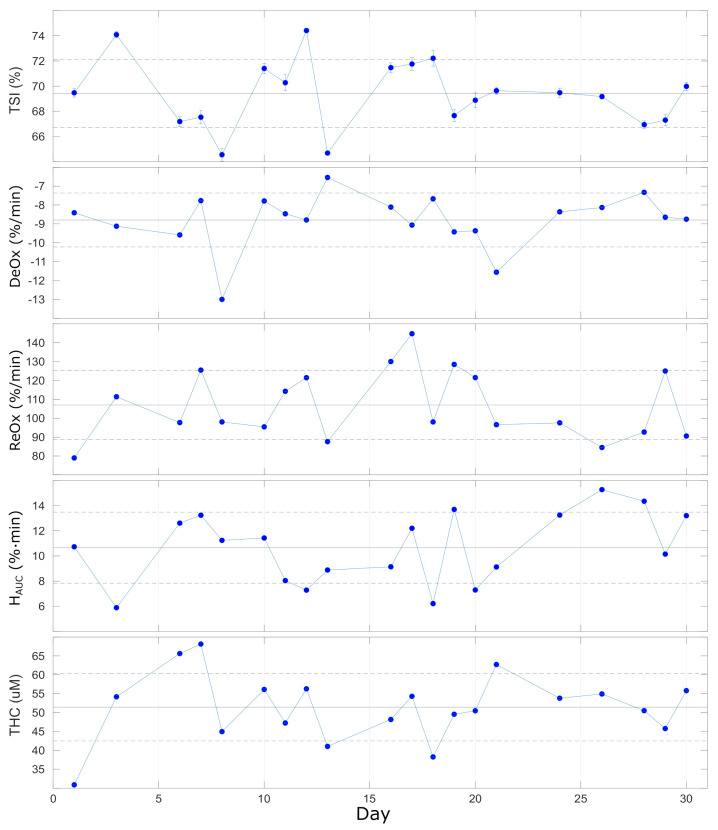
Parameters extracted from VOT over 20 repeated (once/day) measurements over 30 days. Horizontal lines represent the average (solid line) ± standard deviation (dashed lines) over all the measurements. See text for details.

**Table 5 sensors-21-06957-t005:** Vascular occlusion test on a single healthy subject repeated once per day on 20 days over a month, indicating the average values and the variability. See text for details.

${\mathrm{TSI}}_{\mathrm{baseline}}$	DeOx	ReOx	$H_{\mathrm{AUC}}$	${\mathrm{THC}}_{\mathrm{baseline}}$
(%)	(%/min)	(%/min)	(%· min)	(μ M)
69.4 ± 2.7	−8.8 ± 1.4	107 ± 18	10.7 ± 2.8	51.4 ± 8.9
$CV_{TSI}=$ 4%	$CV_{DeOx}=$ 16%	$CV_{ReOx}=$ 17%	$CV_{H_{AUC}}=$ 26%	$CV_{THC}=$ 17%

## 5. Conclusions

In this paper, we have addressed, through systematic and detailed tests, some of the possible challenges and critical issues related to utilizing multiple CW-NIRS devices for a multi-center clinical trial, such as the HEMOCOVID-19 trial, which spans several months.

By characterizing 13 devices from the same manufacturer that are research-grade devices and are not medical devices, we have observed a very low coefficient of variation for both TSI and THC (Table 2) when considering single acquisitions, acquisitions after probe repositioning, performed on different hours, days, and months, and with the device operated by different operators while using standard tissue-simulating phantoms. In these terms, all devices were capable of performing precise and reproducible measurements. On the other hand, in the same phantoms, we have detected significant differences in absolute values between different devices, which indicates that one should be careful about the accuracy of these devices.

Similar measurements were performed in vivo, on the *brachioradialis* muscle of the forearm, and during a vascular occlusion test. The in vivo measurements confirmed that the variabilities associated with physiological changes and errors in the repositioning of the probe are much larger than the inter-device variability, suggesting that in vivo measurements conducted with different devices are comparable.

Finally, we have characterized a set of 10 nominally identical phantoms that could be used in the future for day-by-day device assessment. These phantoms are suitable for the longitudinal assessment of device stability, but not for the evaluation of accuracy of TSI and THC.

In terms of the goal of this characterization for the HEMOCOVID-19 clinical trial, we have ruled out two devices from the study and have decided that, due the differences registered between different devices, operators should proceed in a statistical manner when analyzing the in vivo data.

In conclusion, the performance assessment tests reported in this paper tried to tackle the common problems of the reliability of comparing absolute measurements (and not only relative changes) performed with different CW-NIRS devices. We have provided suggestions for future improvements and suggest care in the use of these systems in clinical trials.

## Figures and Tables

**Figure 1 sensors-21-06957-f001:**
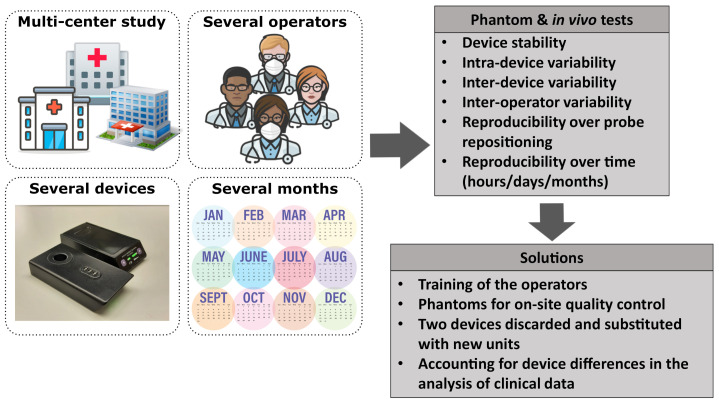
A schematic summary of the challenges of a multi-center clinical study and the actions taken to address them. Icons taken from [29].

**Table 1 sensors-21-06957-t001:** Summary of reported tests. See text for details.

#	Test	Phantom	In Vivo
A	Warm-up and stability	INO PB312	/
B	Variability: single acquisition	INO PB300 & PB312	*Brachioradialis* muscle
C	Reproducibility: probe repositioning	INO PB300 & PB312	*Brachioradialis* muscle
D	Reproducibility over: hours/days/months	INO PB300 & PB312	/
E	Reproducibility: different operators	INO PB300 & PB312	/
F	Differences between devices	INO PB300 & PB312	*Brachioradialis* muscle
G	Type 2 phantom characterization	BioPixS matrix	/
H	Reproducibility: vascular occlusion test	/	*Brachioradialis* muscle

**Table 3 sensors-21-06957-t003:** Type 2 phantoms measured with device id50. See text for details. Values reported for each phantom are average and standard deviation (and corresponding CV) for the three different days of measurement.

Phantom	TSI	${\mathrm{CV}}_{\mathrm{TSI}}$	THC	${\mathrm{CV}}_{\mathrm{THC}}$
	(%)	(%)	(μ M)	(%)
#1	50.6 ± 0.8	1.6	42.4 ± 0.4	0.9
#2	50.2 ± 0.4	0.9	44.1 ± 0.4	1.0
#3	50.6 ± 0.7	1.4	44.5 ± 0.6	1.3
#4	51.7 ± 0.9	1.6	41.6 ± 1.0	2.4
#5	51.5 ± 0.6	1.3	41.3 ± 0.7	1.7
#6	52.7 ± 0.5	0.9	39.9 ± 0.1	0.3
#7	51.2 ± 0.4	0.7	42.5 ± 0.3	0.7
#8	54.5 ± 0.4	0.8	37.2 ± 0.8	2.2
#9	50.4 ± 0.6	1.3	44.6 ± 0.7	1.5
#10	54.2 ± 0.7	1.2	40.9 ± 1.0	2.5

**Table 4 sensors-21-06957-t004:** In vivo variability on *brachioradialis* muscle. See text for details.

Parameter	${\mathrm{CV}}_{\mathrm{single}\;\mathrm{acq}.}$	${\mathrm{CV}}_{\mathrm{replac}.}$	${\mathrm{CV}}_{\mathrm{device}}$
	(%)	(%)	(%)
TSI	0.9	8.5	2.4
THC	2.4	17.5	5.7

## Data Availability

Data are available from the authors upon reasonable request.

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
