# Peer review of "Performance Assessment of a Commercial Continuous-Wave Near-Infrared Spectroscopy Tissue Oximeter for Suitability for Use in an International, Multi-Center Clinical Trial"

_sensors, 2021, doi:10.3390/s21216957_

Round 1
Reviewer 1 Report
This paper investigates the continuous-wave near-infrared method to apply on international, multi-center clinical trial. In the review, I think the author should make major revision to discuss more detail on some key technology.
- The abstract is too lengthy, it should be shorten and respond to the result of this paper.
- The equipment of Fig. 2(experimental setup) should give the type and manufacturer. Furthermore, the author should describe the function of this equipment.
- The author should describe the relationship of the continuous-wave near-infrared and the equipment.
- The author should give scientific explanations on the results. And the authors should give the description how your research is related to sensors.
Author Response
We thank the Reviewer for their comments, here below our answer and changes in the manuscript.
Comment 1:
We agree with the Reviewer and we have reduced the abstract length, reducing the introductory part.
Comment 2:
Action: We have added information about type and manufacturer in Figure 2 caption.
Comment 2 & 3:
CW-NIRS is a well-established technique for local tissue oxygenation monitoring. A full detailed description of the technology and theoretical background falls outside the scope of this paper. In the manuscript we have reported a short but consistent description of the technology, with references to papers describing the technology in detail.
The devices used, PortaMon (Artinis Medical Systems), are commercially available CW-NIRS devices. As reported in Section 3.1, they retrieve the tissue oxygenation by using the spatially resolved spectroscopy and the modified Beer-Lambert law. Unfortunately, the exact algorithms are not available since they are proprietary technology.
Action: We have added two references (ref. [30] and [31]) to Artinis webpages about PortaMon and the technology behind.
Comment 4:
We thank the reviewer for the comment.
To improve the comments about the results obtained we have added:
i) A sentence in Section 4.2: Lastly, we have detected significant differences between different devices (CV<5%) which was significantly larger than all the intra-device CV in phantoms. These differences can be ascribed to device components (i.e. detector, laser, electronics), their assembly and manufacturer internal calibration.
ii) A sentence in section 4.4: These findings once more highlight the difficult of obtaining reliable absolute measurements using CW-NIRS devices. This may be remedied by an intensive training of the operators, better engineering of the device-tissue interface (``probe'') […]
About the second point raised by the reviewer, we note that the manuscript has been submitted to MDPI Sensors special issue on “Biomedical Sensing Applications of Diffuse Optics”, that is the natural location for CW-NIRS technology.
To avoid confusion, we have added the key-word “sensor” in combination with CW-NIRS in the abstract and in the introduction of the paper:
Abstract:
Despite the wide range of clinical and research applications, the reliability of continuous wave near-infrared spectroscopy sensors for absolute oxygenation measurements […]
Introduction:
[…] Despite this variability, the use of CW-NIRS sensors for tissue oxygenation monitoring is well accepted […]
Reviewer 2 Report
This study investigates the stability and variability of the data obtained by continuous-wave near-infrared spectroscopy (CW-NIRS) in a multi-center clinical trial. The authors measured the device stability, intra-device variability, inter-device variability, inter-operator variability, reproducibility over probe positioning, reproducibility over occasions and 0n-site quality control by obtaining the data in phantoms and subjects. The results showed differences between these different conditions.
This is an interesting study investigating the reliability of a certain type of CW-NIRS device. It would be interesting to describe the possible mechanisms of the differences tat the authors observed in the study.
Author Response
We thank the reviewer for revising the manuscript and finding our work “interesting”.
We comment here that, while we have seen that the intra-device variability is mainly affected by the uncertainty in repositioning the device over the same location, the inter-device differences can be also ascribed to the hardware.
To make the point clearer, we have added a sentence in Section 4.2: “Lastly, we have detected significant differences between different devices (CV<5%) which was significantly larger than all the intra-device CV in phantoms. These differences can be ascribed to device components (i.e. detector, laser, electronics), their assembly and manufacturer internal calibration.”
Reviewer 3 Report
In this paper, the authors have evaluated the suitability of CW-NIRS devices for possible use in multi-centre-international clinical trials. This is the most important issue in using non-invasive devices, which authors have investigated. Overall, the paper is well written and well structured. A detailed description of tests and validations are provided. I would like to appreciate the efforts of the authors to highlight and provide insight on this critical issue. I would like to accept this manuscript with the following minor observations.
- Figure 1 is not convincing. Figure 1 should be improved to depict the importance of this study
- There is some problem with Figure 2. I am unable to see Figure 2. Kindly check this issue before submitting the revised manuscript.
- I would suggest including a figure showing a comprehensive methodology for a better understanding of the readers.
Author Response
We thank the reviewer for their comments and for considering our manuscript ready for publication after addressing their comments
Comment 1: Following the reviewer’s suggestion, we have changed Figure 1, highlighting the action taken in light of the results of the tests reported in the manuscript.
Comments 2 & 3: We are sorry that the reviewer was not able to display Figure 2. The figure is present in the manuscript, and we will double check before publication. With respect to comment 3, Figure 2 indeed shows the experimental setup and methodology.
Reviewer 4 Report
This is an excellent study by Cortese et. al. to look at the reliability of near-IR based absolute oxygenation measuring instruments that in use for HEMOCOVID-19 clinical trials. This is very well controlled study with range of instruments that are presently in use. The experimental design and conclusions are very pertinent and is of critical interest to everyone during this pandemic. The manuscript is ready to accepted in it's present form.
Author Response
We thank Reviewer 4 for their positive evaluation, and for considering the manuscript ready for publication in its present form.
Round 2
Reviewer 1 Report
All comments have modified and replied. The paper could be accepted as this revised form.